# Supercritical CO$_2$ Exposure-Induced Surface Property, Pore Structure, and Adsorption Capacity Alterations in Various Rank Coals

**Zhenjian Liu** [1,2,3,*]**, Zhenyu Zhang** [2,3] 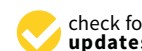**, Xiaoqian Liu** [2,3]**, Tengfei Wu** [4,5] **and Xidong Du** [6]

1   College of Civil Engineering, Yancheng Institute of Technology, Yancheng 221051, China
2   State Key Laboratory of Coal Mine Disaster Dynamics and Control, Chongqing University, Chongqing 400044, China
3   Geofluids, Geomechanics and Geoenergy (3G) Research Group, Chongqing University, Chongqing 400044, China
4   China Coal Technology and Engineering Group Shenyang Research Institute, Fushun 113122, China
5   State Key Laboratory of Coal Mine Safety Technology, Fushun 113122, China
6   School of Earth Sciences, East China University of Technology, Nanchang 330013, China
*   Correspondence: zjliu@cqu.edu.cn; Tel.: +86-188-7520-8316

**Abstract:** Carbon dioxide (CO$_2$) has been used to replace coal seam gas for recovery enhancement and carbon sequestration. To better understand the alternations of coal seam in response to CO$_2$ sequestration, the properties of four different coals before and after supercritical CO$_2$ (ScCO$_2$) exposure at 40 °C and 16 MPa were analyzed with Fourier Transform infrared spectroscopy (FTIR), low-pressure nitrogen, and CO$_2$ adsorption methods. Further, high-pressure CO$_2$ adsorption isotherms were performed at 40 °C using a gravimetric method. The results indicate that the density of functional groups and mineral matters on coal surface decreased after ScCO$_2$ exposure, especially for low-rank coal. With ScCO$_2$ exposure, only minimal changes in pore shape were observed for various rank coals. However, the micropore specific surface area (SSA) and pore volume increased while the values for mesopore decreased as determined by low-pressure N$_2$ and CO$_2$ adsorption. The combined effects of surface property and pore structure alterations lead to a higher CO$_2$ adsorption capacity at lower pressures but lower CO$_2$ adsorption capacity at higher pressures. Langmuir model fitting shows a decreasing trend in monolayer capacity after ScCO$_2$ exposure, indicating an elimination of the adsorption sites. The results provide new insights for the long-term safety for the evaluation of CO$_2$-enhanced coal seam gas recovery.

**Keywords:** super-critical CO$_2$; coal rank; surface property; pore structure; adsorption

## 1. Introduction

Recently, concerns about increasing carbon dioxide (CO$_2$) concentration in the atmosphere have driven research into the technical reduction of the emission of CO$_2$ from fossil fuel use [1]. Carbon capture and storage (CCS) is an effective approach for CO$_2$ mitigation currently under consideration [2,3]. Deep, un-minable coal seams have been targeted as safer sites for CO$_2$ sequestration because the sequestrated CO$_2$ is predominantly stored as a relatively stable adsorption phase in coal seams. Meanwhile, the replacement of coal seam gas (CSG) offsets some proportion of sequestration costs [4–6]. Accordingly, an understanding of the physical and chemical alterations of coal reservoirs in response to CO$_2$ sequestration would offer a scientific foundation on which to base long-term storage predictions.

Recently, several studies have investigated the chemical and structural properties of various rank coals [7–10]. These studies have demonstrated that coal mass is an organic-rich porous rock, consisting of micropores (<2 nm in diameter) and mesopores (2–50 nm in diameter) in coal matrix and cleats (>50 nm in diameter). The structural properties of coal vary with the degree of maturity as determined by reflectance or volatile matter [11]. In the $CO_2$-enhanced coal bed methane recovery ($CO_2$-ECBM) process, $CO_2$ fluids first flow into the cleat system of coal mass and then slowly diffuse into mesopores and micropores. In mesopores and micropores, $CO_2$ can displace the pre-adsorbed CSG through the competitive adsorption mechanism, thus achieving $CO_2$ storage and CSG enhancements [12].

The swelling effect of coal matrix induced by $CO_2$ adsorption has been reported extensively [13–16]. As reviewed by Perera et al. [17], the swelling of the coal matrix is accompanied by a series of irreversible effects on coal seam, including the variation of coal seam permeability and mechanical properties. The matrix swelling effect is closely related to coal rank, where higher-ranked coal exhibits a lower matrix swelling rate [18]. Normally, the potential coal seams for $CO_2$ sequestration are at depths of 800–1000 m, where temperature and pressure are beyond the critical point of $CO_2$ ($T_c$ = 31.8 °C, $P_c$ = 7.38 MPa) and $CO_2$ exists in its supercritical phase. Supercritical $CO_2$ ($ScCO_2$) exhibits liquid-like density and dissolution power as well as gas-like viscosity and transport behavior [19]. Laboratory experiments have demonstrated that $ScCO_2$ adsorption causes a significantly higher coal matrix swelling effect in comparison with subcritical $CO_2$ adsorption due to the higher uptake capacity of $ScCO_2$ [6,20]. Further, Hol et al. [21] revealed that the adsorption-induced heterogeneous swelling at the maceral scale, accompanied by differential accessibility of the coal microstructure, can form microfractures in the coal mass. In this respect, the sequestrated $CO_2$ may alter the pore morphology of the coal mass during long-term storage.

In addition to the adsorption-induced coal matrix swelling effect, $CO_2$ molecules can dissolve into the coal matrix, especially when $CO_2$ exists in a supercritical state [22,23]. According to Goodman et al. [24], $CO_2$ is a more chemically potential adsorbate than CSG and can act as plasticizer of coal mass, leading to a potential structural rearrangement that affects the molecular structure or intermolecular bonding of coal matrix. Moreover, $ScCO_2$ can act as a solvent capable of extracting organic hydrocarbons from the coal matrix as demonstrated by Kolak and Burruss [25]. Their experiments demonstrated that the extracted amount and type of hydrocarbons varied for different rank coals, with the highest hydrocarbon concentrations mobilized from high volatile bituminous coal.

In consideration of the effects of the fluid-solid interaction between $ScCO_2$ and coal matrix, it is essential to evaluate the chemical and structural alterations in coal with $ScCO_2$ exposure. The alterations in coal properties may further change the flow behavior and adsorption capacities of $CO_2$ on the coal mass, thus influencing the substantial $CO_2$ injection rate and CSG recovery efficiency. Previous studies only considered the effects of $ScCO_2$ on the chemical and structural properties of the coal mass. Systematic knowledge of the adsorption capacity alteration of various rank coals with $ScCO_2$ exposure is still needed to better estimate the feasibility of $CO_2$ sequestration in coal seams.

The objective of this study is to investigate the variation of chemical and structural properties as well as the corresponding alteration of the adsorption capacity in response to $ScCO_2$ exposure for various rank coals. To address this issue, Fourier transform infrared spectroscopy (FTIR) analysis, low-pressure $N_2$ and $CO_2$ sorption method, and high-pressure $CO_2$ adsorption experiments were conducted on four Chinese coals of different rank.

## 2. Materials and Methods

### 2.1. Sample Collection and Preparation

Four coal samples collected from coal mines in China were used in this study: lignite from No. 1–2 coal seam in Daliuta coal mine, Shendong coal field (DL coal); middle volatile bituminous coal from No. 2 coal seam in Zhaolou coal mine, Heze coalfield (ZL coal); low volatile bituminous coal from No. C-9 coal seam in Guiyuan coal mine, Guizhou coalfield (GY coal); anthracite from No. 6 coal seam in

Datong coal mine, Sichuan coalfield (DT coal). The information of the sampling coal mines is shown in Figure 1.

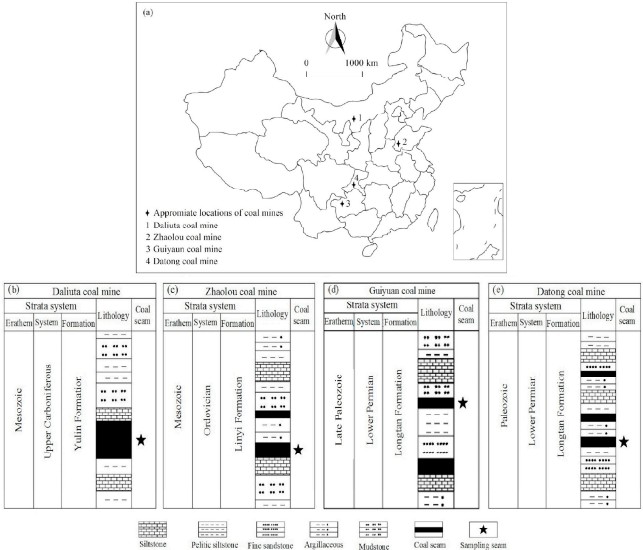

**Figure 1.** The information of the sampling coal mines: (**a**) location, and (**b**–**e**) stratigraphic column.

Coal blocks were first drilled from freshly exposed mining faces and then immediately sealed in a plastic bag and transported to the laboratory. To accelerate the coal–$ScCO_2$ interaction, all the coal lumps were crushed and sieved to particles of 18–20 mesh. Proximate and ultimate analyzes were carried out to characterize coal samples. The analysis results are presented in Table 1.

**Table 1.** Proximate and ultimate analyzes of the coal samples.

| Sample | $R_{o\,max}$ (%) | Proximate Analysis (wt %) | | | | Ultimate Analysis (wt % daf) | | | |
|--------|------------------|-----------|-----------|-----------|------|------|------|------|------|
| | | $C_{fix}$ | $V_{daf}$ | $A_{ad}$ | $M$ | $C$ | $H$ | $N$ | $O$ |
| DL coal | 0.42 | 53.55 | 36.96 | 19.42 | 2.46 | 72.71 | 4.95 | 1.19 | 20.50 |
| ZL coal | 0.81 | 65.10 | 28.71 | 16.24 | 1.50 | 82.54 | 4.57 | 1.08 | 11.03 |
| GY coal | 1.14 | 78.34 | 6.14 | 10.52 | 2.08 | 86.92 | 4.01 | 1.20 | 6.03 |
| DT coal | 1.86 | 73.85 | 12.81 | 13.34 | 1.96 | 90.11 | 3.79 | 0.96 | 2.05 |

Notes: $C_{fix}$, fixed carbon; $V_{daf}$, volatile matters; $A_{ad}$, ash; $M$, moisture.

## 2.2. Exposure of Coal to ScCO₂

In this study, the raw coal samples were firstly saturated to $ScCO_2$ and then further tests were conducted to compare coal properties alteration before and after $ScCO_2$ exposure. A specific high-pressure sealed vessel was fabricated and used to saturate the coal samples with $ScCO_2$ as schematically shown in Figure 2. The heart of this apparatus are pressure cells, an ISCO (260D) syringe pump (Teledyne ISCO, Lincoln, NE, USA) and the data system. The sample was first inserted into the pressure cell with a temperature-controlled thermostatic water bath with an accuracy of ± 0.1 °C. Then, $CO_2$ was injected from the gas cylinder into the sample cell by the ISCO syringe pump. Prior to interaction, the system was degassed under vacuum at 60 °C for 12 h. The reaction pressure was set as 16 MPa, with a constant temperature value of 40 °C, simulating $CO_2$ injection into deep (~800 m) coal seams [17]. The exposure time was set at 30 days to ensure a sufficient coal–$ScCO_2$ interaction. After the saturation period, the sample cell was gradually de-pressurized by 0.1 MPa/min to avoid possible damage to the coal samples. The coal samples were then covered with plastic wrap to avoid any possible changes to the saturation state. All the tests were repeated three times to increase the confidence level and reliability. The average data of three tests were used in this study.

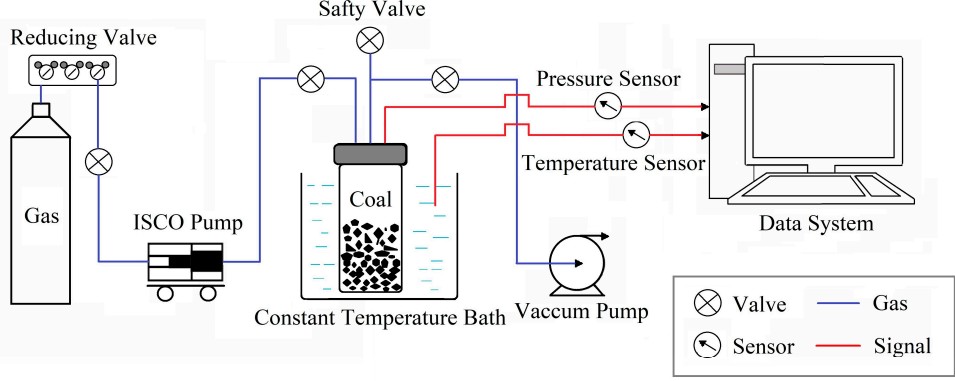

**Figure 2.** Schematic diagram of coal–ScCO$_2$ interaction system.

### 2.3. Fourier Transform Infrared Spectroscopy (FTIR) Analysis

The surface functional groups of both the raw and ScCO$_2$-treated coal samples were determined using a FTIR Nicelet 6700 (Thermo Fisher Scientific, Waltham, MA, USA). The coal samples for FTIR analysis were prepared following the potassium bromide (KBr) pellet method. The coal powders (0.5–1 mg in weight) and the dried KBr were ground at a mass ratio of 1:160. The spectral was obtained in the 400 cm$^{-1}$ and 4000 cm$^{-1}$ wavenumber regions with a resolution of 4 cm$^{-1}$.

### 2.4. Low-Pressure N$_2$ and CO$_2$ Isotherm Measurements

Low-pressure gas adsorption measurements were conducted on an Accelerated Surface Area and Porosimeter System ASAP 2020M (Micrometrics Instruments Corporation, Norcross, GA, USA). Using standard adsorbates N$_2$ and CO$_2$, information about the mesopore and micropore structure of the coal samples was obtained. Before analysis with either N$_2$ or CO$_2$, the samples (weighing 1–2 g) were degassed under vacuum by heating at 105 °C for 12 h to remove water and other volatile matter. The adsorption-desorption isotherms of N$_2$ were measured at liquid nitrogen (−196.15 °C) under the relative pressure ranging from 0.005 to 0.99. The isotherms of CO$_2$ adsorption were performed at 0 °C under the relative pressure of 0.005–0.032.

At each pressure set point, the sorption equilibrium was established automatically when the system pressure remained stable for 30 s. The absolute pressure tolerance was set as 5 mmHg (6.66 mbar). After each test, the specific surface area (SSA), pore volume, and pore size distribution (PSD) of the coal samples were calculated based on multiple models.

### 2.5. High-Pressure CO$_2$ Isotherm Measurements

High-pressure CO$_2$ adsorption isotherms were performed using a gravimetric analyzer IGA (Hiden Isochema Limited, Warrington, UK). A detailed description of the IGA system and the general adsorption isotherm test procedure can be found in reference [26]. All isotherms were measured to a pressure of 1.8 MPa at 40 °C. Prior to tests, the samples were degassed at 105 °C under vacuum (<10$^{-6}$ Pa) for 12 h to remove the water content from the samples. During the experiment, equilibrium was established when the rate of sample weight change was less than 1% or a time limit was reached. Then, the gas pressure was regulated to the next step. The buoyancy effect was corrected automatically in the measurement.

## 3. Results and Discussion

### 3.1. Effect of ScCO$_2$ Exposure on Coal Surface Property

The effect of ScCO$_2$ exposure on surface property of various rank coals was examined by FTIR spectroscopy, a technique well suited to study the surface functionalities. The FTIR spectra for the raw and ScCO$_2$-treated coal samples are shown in Figure 3. As presented in Figure 3, the FTIR spectra of

the coal samples were typical of complex, heterogeneous organic materials containing multiple type of surface functional groups. With $ScCO_2$ exposure, both the -OH groups (at a wave number of around 3450 cm$^{-1}$) and the amide carbonyl groups (at a wave number of around 1620 cm$^{-1}$) show an obvious decrease for DL coal, ZL coal, and GY coal, with only a slight decrease for DT coal. The prominent bands at the wave number around 1000 cm$^{-1}$ possibly contributed to mineral matter and clay, which show an obvious decrease for all the coals after $ScCO_2$ exposure, indicating an elimination of mineral matter and clay. The bands ranging 900–400 cm$^{-1}$ show an obvious decrease for DL coal and DT coal after $ScCO_2$ exposure, which is attributed to the Si-O and Si-O-Al bands.

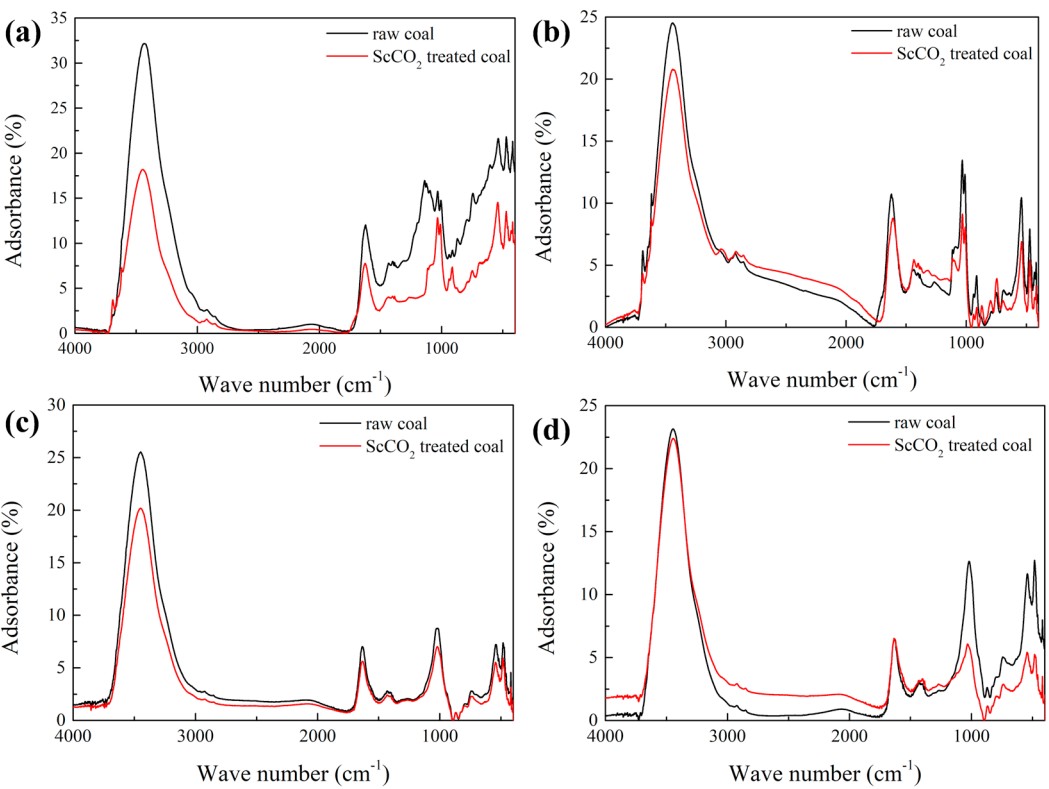

**Figure 3.** Fourier transform infrared spectroscopy (FTIR) spectra for the raw and $ScCO_2$-treated coal samples: (**a**) DL coal; (**b**) ZL coal; (**c**) GY coal; (**d**) DT coal.

In a comparison of the spectra of the raw and $ScCO_2$-treated coal samples, obvious variations were found in functional group distribution, especially for the low-rank coal. According to Wang et al. [27], the decrease in surface functional group density may be attributable to the extracting effect or reactivity of $ScCO_2$ fluid. The decrease of -OH groups may be related to the dissolution of $ScCO_2$ to the inherent moisture in the coal samples; however, the elimination of the amide carbonyl groups may be related to the extraction of the organic volatile corresponds from coal with $ScCO_2$ exposure, as demonstrated by Li et al. [28]. In addition, a decrease in the distribution of mineral matter content in the coal samples was observed after $ScCO_2$ exposure. However, Mastalerz et al. [29] detected no change in the functional groups of two different coal samples after saturation with $CO_2$. This is because the saturation conditions they conducted were at 20 °C and 4.14 MPa where $CO_2$ was in a sub-critical state in which it cannot extract corresponds from coal matrix. The loss of the organic and inorganic corresponds in the coal matrix with $ScCO_2$ exposure may influence its adsorption capacity for $CO_2$, which will be discussed later in this study.

### 3.2. Effect of ScCO₂ Exposure on Coal Pore Structure

### 3.2.1. Low-Pressure N₂ and CO₂ Sorption Isotherms

The adsorption-desorption isotherms of $N_2$ on the raw and $ScCO_2$-treated coal samples at $-196$ °C are shown in Figure 4. All the isotherms correspond to the type IV isotherm of IUPAC classification [30]. This type of isotherm is considered to be associated with the distribution of mesopores in solids [31]. Distinct adsorption-desorption hysteresis loop is observed in each isotherm at a high relative pressure (>0.45). The presence of the hysteresis loop suggests that capillary condensation occurred within the mesopores. The shape of the hysteresis loop is a reflection of pore morphology within the coal matrix [32]. As shown in Figure 4, there is only a slight difference in the shape of the hysteresis loop in the isotherms between the raw and $ScCO_2$-treated coal samples. It can be deduced that interaction with $ScCO_2$ has a minimal influence on pore shape in various rank coals. This is consistent with previous studies [33,34].

Figure 4 also shows that the adsorbed amount of $N_2$ on the $ScCO_2$-treated coal samples is lower than that on the raw samples, suggesting that the adsorption capacity of $N_2$ on various rank coals uniformly decreased after $ScCO_2$ exposure. At relative pressure less than 0.2, there is no distinct difference in $N_2$ adsorption capacity between the raw and $ScCO_2$-treated coal samples. With the elevated relative pressure, the adsorption amount for the raw coal samples increases more rapidly than the $ScCO_2$-treated coal samples. This implies that exposure to $ScCO_2$ mainly influences the multilayer capacity of $N_2$ adsorption. In this respect, exposure to $ScCO_2$ may have a more distinct effect on the broader mesopores than on the narrow mesopores.

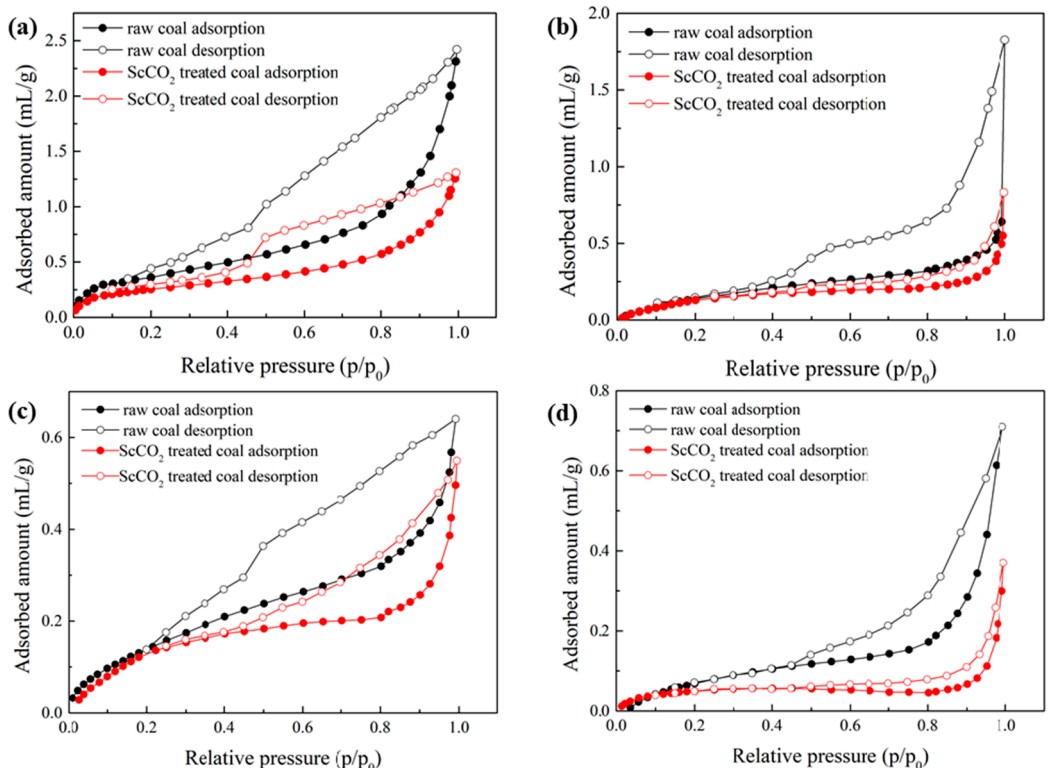

**Figure 4.** $N_2$ adsorption-desorption isotherms on the raw and $ScCO_2$-treated coal samples at $-196$ °C: (**a**) DL coal; (**b**) ZL coal; (**c**) GY coal; (**d**) DT coal.

The $N_2$ adsorption-desorption method can only provide mesopore information of the coal samples because $N_2$ molecules cannot penetrate the fine micropores and pore shrinkage of coal at the extremely low temperature of $-196.15$ °C. Low pressure $CO_2$ adsorption at 0 °C can overcome this drawback due to the shorter equilibrium time and disappearance of the pore shrinkage effect [35].

Figure 5 shows the adsorption isotherms of $CO_2$ on the raw and $ScCO_2$-treated coal samples at 0 °C under a low relative pressure (≤0.033). A comparison of Figures 4 and 5 reveals that, for various rank coals, the adsorption capacity of $CO_2$ is larger than $N_2$ at the same relative pressure due to the accessibility of $CO_2$ molecules to the narrow micropores of coal matrix in comparison with $N_2$ molecules. Figure 5 shows that the low-pressure adsorption capacity of $CO_2$ on the $ScCO_2$-treated coal samples is higher than that on the raw coal samples. This observation suggests an increase of accessibility in the micropores after $ScCO_2$ exposure, potentially attributable to the micro-fracturing of the coal matrix with $ScCO_2$ exposure, as demonstrated by Hol et al. [21].

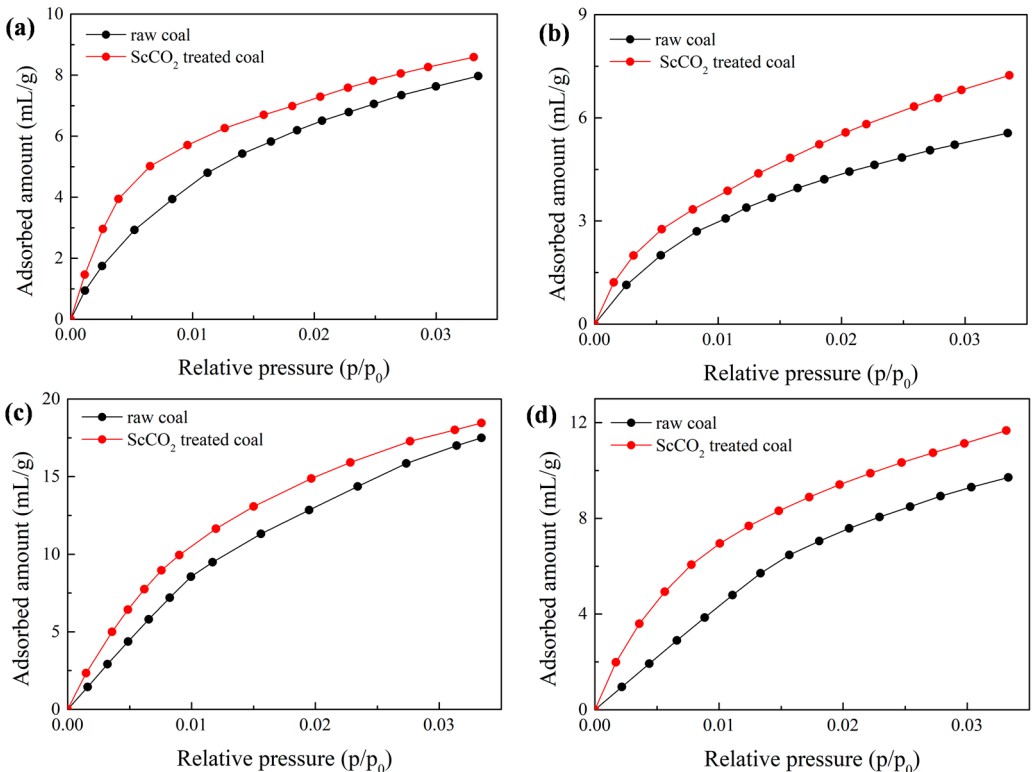

**Figure 5.** Low-pressure $CO_2$ adsorption isotherms on the raw and $ScCO_2$-treated coal samples at 0 °C: (**a**) DL coal; (**b**) ZL coal; (**c**) GY coal; (**d**) DT coal.

### 3.2.2. Pore Properties Determined by Low-Pressure $N_2$ and $CO_2$ Sorption

The mesopore specific surface area (SSA) and total pore volume ($V_t$) of the coal samples with $N_2$ adsorbate were evaluated by the Brunauer-Emmett-Teller (BET) and Barrett-Joyner-Halenda (BJH) models, respectively. The SSA and micropore volume ($V_{mic}$) with $CO_2$ adsorbate were evaluated by Dubinin-Radushkevich (D-R) and Dubinin-Astakhov (D-A) models [27]. These data can be calculated automatically by the analytical software in ASAP2020.

The estimated SSA and pore volume of the raw and $ScCO_2$-treated coal samples from $N_2$ and $CO_2$ sorption data are summarized in Table 2. The results show that, with $ScCO_2$ exposure, the mesopore SSA and $V_t$ of the coal samples are slightly decreased, as seen in Figure 6. As shown in Table 2, reductions of about 0.366 to 0.065 $m^2/g$ of mesopore SSA and $9 \times 10^{-3}$ to $2 \times 10^{-3}$ $cm^3/g$ of $V_t$ values can be observed after $ScCO_2$ exposure. These results indicate that the accessibility of mesopore in various rank coals decreased after $ScCO_2$ exposure. This may have been induced by the irreversible swelling effect of the coal matrix, which may block the pathways of gas molecules and reduce the pore volume [36]. With regard to the average pore width ($D$), the average pore diameter of GY coal reduced sharply after $ScCO_2$ exposure while the values for DL coal, ZL coal and NM coal increased

slightly. This may be due to the larger matrix swelling of GY coal by ScCO$_2$ exposure than the other coal samples.

Unlike mesopore, the values of micropore SSA and $V_{mic}$ for the coal samples increased notably with ScCO$_2$ exposure, as seen in Figure 7. According to Table 2, the D-R SSA ($S_{D-R}$) and D-A SSA ($S_{D-A}$) determined by CO$_2$ adsorption exhibit a similar trend with a 15.8 to 137.0 m$^2$/g and 19.3 to 44.2 m$^2$/g increase for various rank coals, respectively. Simultaneously, increases of 0.005 to 0.014 cm$^3$/g of $V_{mic}$ values can be observed after ScCO$_2$ exposure. The increased amount of micropore SSA and $V_{mic}$ in GY coal is the largest, followed by DT coal, DL coal, and ZL coal, respectively.

**Table 2.** Structural properties of the raw and ScCO$_2$-treated coal samples determined by N$_2$ and CO$_2$ sorption.

| Sample | State | N$_2$ Adsorption | | | CO$_2$ Adsorption | | |
|---|---|---|---|---|---|---|---|
| | | $S_{BET}$/m$^2 \cdot$g$^{-1}$ | $V_t$/cm$^3 \cdot$g$^{-1}$ | $D$/nm | $S_{D-R}$/m$^2 \cdot$g$^{-1}$ | $S_{D-A}$/m$^2 \cdot$g$^{-1}$ | $V_{mic}$/cm$^3 \cdot$g$^{-1}$ |
| DL coal | raw coal | 1.498 | 0.0068 | 18.16 | 98.71 | 61.05 | 0.0216 |
| | ScCO$_2$-treated | 1.132 | 0.0060 | 21.20 | 118.5 | 84.18 | 0.0327 |
| ZL coal | raw coal | 0.744 | 0.0043 | 23.12 | 54.54 | 45.77 | 0.0219 |
| | ScCO$_2$-treated | 0.611 | 0.0041 | 26.84 | 92.65 | 65.10 | 0.0268 |
| GY coal | raw coal | 0.616 | 0.0028 | 18.19 | 222.4 | 160.9 | 0.0667 |
| | ScCO$_2$-treated | 0.551 | 0.0019 | 13.79 | 359.4 | 205.1 | 0.0810 |
| DT coal | raw coal | 0.586 | 0.0016 | 10.92 | 120.4 | 74.34 | 0.0264 |
| | ScCO$_2$-treated | 0.387 | 0.0011 | 11.37 | 176.6 | 100.6 | 0.0353 |

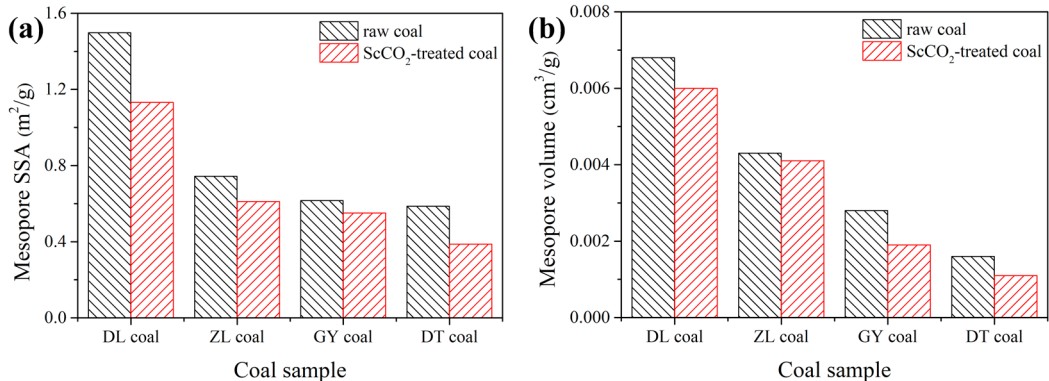

**Figure 6.** Mesopore specific surface area (SSA) and volume of the raw and ScCO$_2$-treated coal samples: (**a**) mesopore SSA; (**b**) mesopore volume.

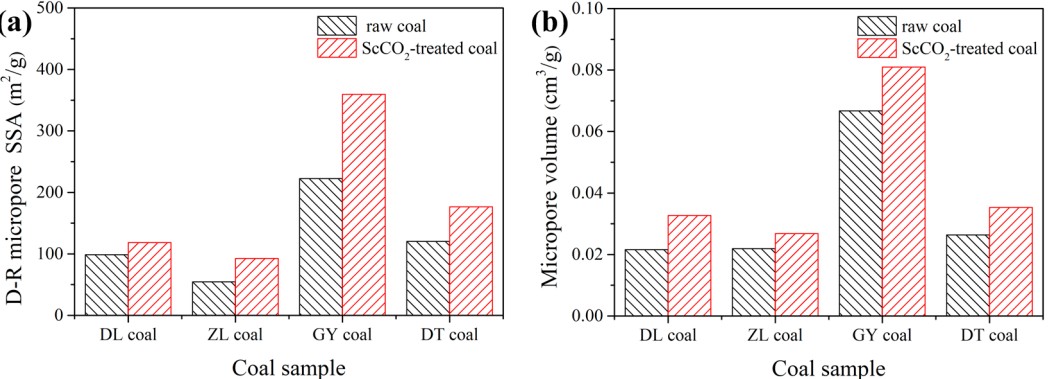

**Figure 7.** D-R micropore SSA and volume of the raw and ScCO$_2$-treated coal samples: (**a**) micropore SSA; (**b**) micropore volume.

In combination with the $N_2$ and $CO_2$ adsorption results, the alterations of the pore structure in various rank coals were mainly controlled by two distinct mechanisms: (1) the extraction effect of $ScCO_2$ removes some volatile matters in the pore medium; (2) $S_CCO_2$ exposure induces irreversible swelling which may partially block the pore throat in the coal medium. The two mechanisms account for the formation of micropore at the expense of mesopore in the coals after $S_CCO_2$ exposure. An increased amount of the micropore is larger than the decreased amount of the mesopore, implying that the combined influence of the two mechanisms results in an increased amount of the adsorptive pore.

### 3.2.3. Pore Size Distribution Determined by Low-Pressure $N_2$ and $CO_2$ Sorption

The mesopore volume distribution of the raw and $ScCO_2$-treated coal samples derived from BJH model are presented in Figure 8. The results indicate that the PSD of raw and $ScCO_2$-treated coal samples show multimodal distribution in the mesopore range. According to Figure 8, contact with $ScCO_2$ does not change the profile of PSD significantly in the mesopore range for various rank coals. However, the major peaks for ZL, GY, and DT coal were weakened after $ScCO_2$ exposure. This observation is consistent with Gathitu et al. [37], who revealed that the mesopores in coal matrix may collapse during the interaction with $ScCO_2$, resulting in a decrease of the mesopore amount in the coal mass. According to Figure 8, pore size distributions are very similar for GY and DT coal at >5 nm, a potential effect of the similar ranks of GY and DT coal.

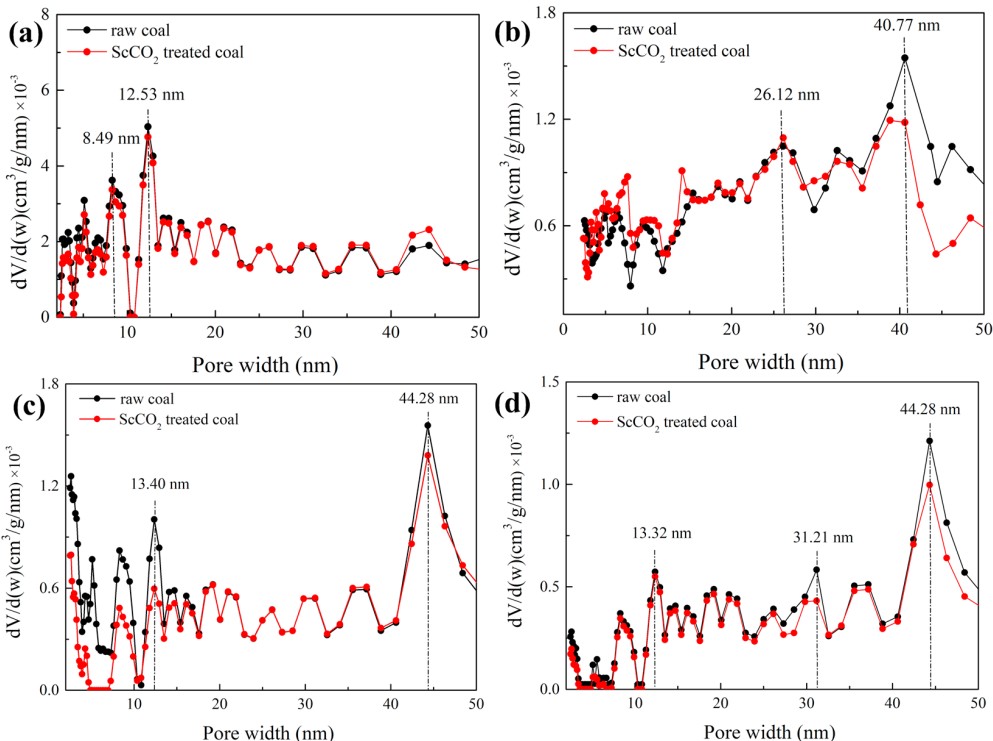

**Figure 8.** Pore size distribution (PSD) for the raw and $ScCO_2$-treated coal samples determined by $N_2$ adsorption: (**a**) DL coal; (**b**) ZL coal; (**c**) GY coal; (**d**) DT coal.

The micropore size distribution of the raw and $ScCO_2$-treated coal samples was determined by nonlocal density functional theory (NLDFT), an effective method to calculate pore size 0.5–1 nm; results are shown in Figure 9. In contrast with a similar mesopore size distribution before and after $ScCO_2$ exposure, the micropore size distribution differs between the raw and $ScCO_2$-treated coal samples, indicating a change in micropore morphology after $ScCO_2$ exposure. The pore width of the raw coal samples shows unimodal distribution with major peaks at 0.80–0.88 nm; in contrast, the pore width of $ScCO_2$-treated coal samples shows multimodal distribution with major peaks at around 0.59–0.65 nm, 0.75–0.8 nm, and 0.82–0.85 nm. This implies that a new micropore formed in various

rank coals with ScCO$_2$ exposure. As mentioned above, this can be interpreted by micro-fracturing of the coal matrix with ScCO$_2$ exposure. Pan et al. [38] further indicated that many closed pores in the raw coals opened and transmitted to adsorption pore with ScCO$_2$ exposure, resulting in an increase of micropore connectivity.

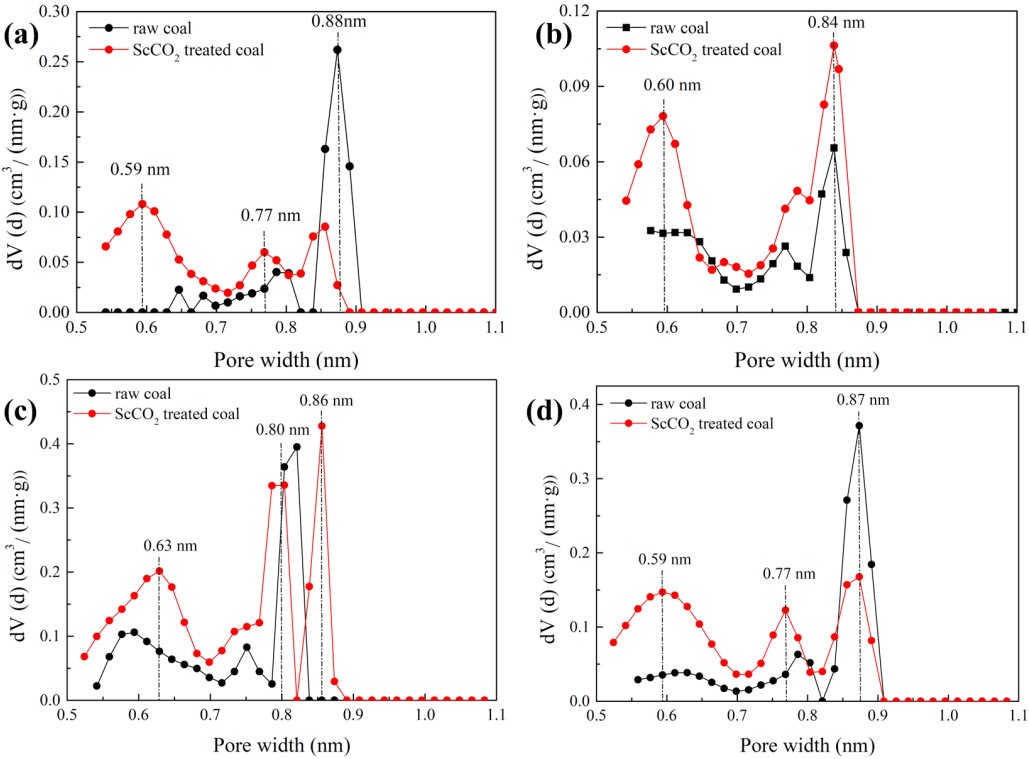

**Figure 9.** PSD for the raw and ScCO$_2$-treated coal samples determined by CO$_2$ adsorption: (**a**) DL coal; (**b**) ZL coal; (**c**) GY coal; (**d**) DT coal.

### 3.3. Effect of Sc-CO$_2$ Exposure on High-Pressure CO$_2$ Adsorption

The high-pressure adsorption isotherms of CO$_2$ on the raw and ScCO$_2$-treated coal samples are shown in Figure 10. All the isotherms belong to type I, indicating an occurrence of monolayer adsorption [30]. Figure 10 shows that, although ScCO$_2$ exposure does not change the shape of the adsorption isotherms for CO$_2$ adsorption, it significantly affects the adsorption capacities of various rank coals. Notably, ScCO$_2$ exposure can enhance CO$_2$ adsorption at a low pressure; however, the CO$_2$ adsorption capacity for all the ScCO$_2$-treated samples is lower than the raw coal samples at higher pressure. This may be the combined effects of surface property and pore structure alterations for the coal samples with ScCO$_2$ exposure. At a low pressure, the adsorption capacity is mainly dominated by the micropore; the increase of micropore volume leads to higher adsorption capacity after ScCO$_2$ exposure. The adsorption transferred to the mesopores at a higher pressure whereas a decrease in mesopore volume resulted in lower adsorption capacity. In addition, the density of surface functional groups on coal surface can promote CO$_2$ adsorption [39,40]. Thus, the extraction of functional groups on the coal surface by ScCO$_2$ may be another reason for the decrease of CO$_2$ adsorption capacity at a higher pressure.

To describe the high-pressure adsorption isotherms of CO$_2$ on various rank coals before and after ScCO$_2$ exposure, the Langmuir model, which is based on monolayer adsorption, is employed, which was given as:

$$n = \frac{n_m b p}{1 + b p} \tag{1}$$

where $n$ is adsorbed amount, $p$ is the equilibrium pressure, $n_m$ is monolayer adsorption capacity, also known as Langmuir volume, and $b$ is a constant known as Langmuir constant [41].

The fitting parameters for the isotherms were determined by Origin Lab software and the best-fit results are summarized in Table 3. The adsorption isotherms were well fitted by Langmuir with $R^2$ greater than 0.99 for all cases. The values of $n_m$ for the raw coals are larger than the $ScCO_2$ treated coals, as seen in Figure 11a, indicating a decrease of monolayer ability after $ScCO_2$ exposure. As mentioned above, this may be due to the reduction of the surface functional group density with $ScCO_2$ exposure. Unlike $n_m$, the values of $b$ for the coal samples were increased, as seen in Figure 11b, suggesting that the monolayer ability can be attained at a lower pressure after $ScCO_2$ exposure.

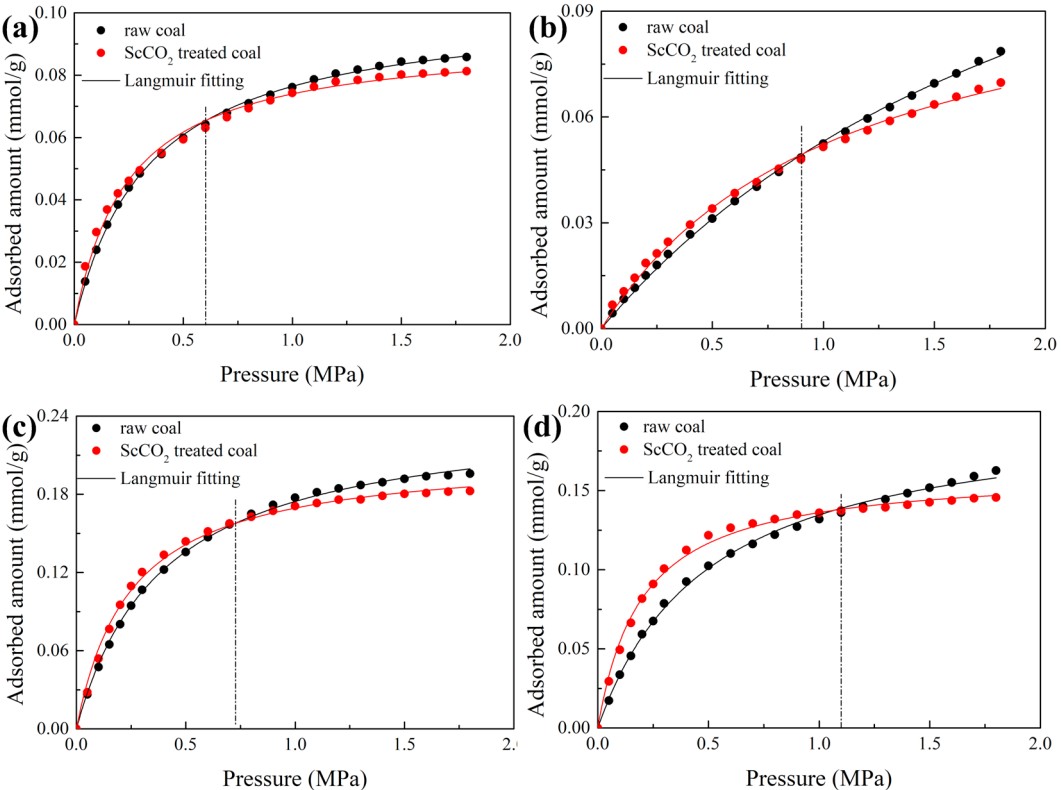

**Figure 10.** Adsorption isotherms of $CO_2$ on the raw and $ScCO_2$-treated coal samples at 40 °C and the fitting curves by Langmuir model: (**a**) DL coal; (**b**) ZL coal; (**c**) GY coal; (**d**) DT coal.

**Table 3.** Fitting parameters of the Langmuir model for $CO_2$ adsorption at 40 °C.

| Sample | State | $n_m$/mmol·g$^{-1}$ | $B$ | $R^2$ |
|--------|-------|---------------------|-----|-------|
| DL coal | raw coal | 0.102 | 2.923 | 0.9993 |
| | $ScCO_2$-treated | 0.092 | 4.056 | 0.9954 |
| ZL coal | raw coal | 0.181 | 0.416 | 0.9991 |
| | $ScCO_2$-treated | 0.110 | 0.906 | 0.9975 |
| GY coal | raw coal | 0.243 | 2.538 | 0.9990 |
| | $ScCO_2$-treated | 0.211 | 4.050 | 0.9964 |
| DT coal | raw coal | 0.202 | 1.994 | 0.9981 |
| | $ScCO_2$-treated | 0.163 | 5.029 | 0.9960 |

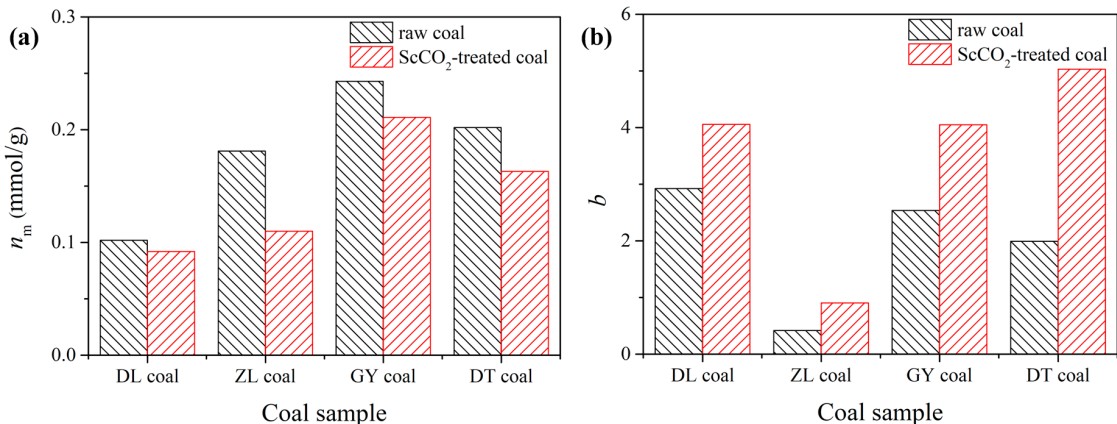

**Figure 11.** Fitting parameters for Langmuir model: (**a**) $n_{\mathrm{m}}$; (**b**) $b$.

*3.4. Implications for $CO_2$ Sequestration in Coal Seams for Coal Seam Gas (CSG) Enhancement*

To date, several projects of ECBM have been carried out in USA, Canada, China, Japan, and Australia. A comprehensive understanding of the effects of $ScCO_2$ exposure on coal seams during the storage is essential to predict the feasibility of these projects [42]. On the basis of the results of this study, $ScCO_2$ will significantly influence the surface property pore structure as well as high-pressure $CO_2$ adsorption capacity in the coal seam during long-term storage. These effects may be aggravated during in-situs conditions, due to the combined effect of $ScCO_2$ and the seam water. Thus, field-scale reservoir tests are still needed to better understand the response of coal seams to $CO_2$ sequestration.

## 4. Conclusions

In this study, the alterations of surface property and pore structure as well as the corresponding high-pressure $CO_2$ adsorption capacity in various rank coals were experimentally studied on raw and $ScCO_2$-treated coal samples ($T = 40\,°C$, $P = 16$ MPa). The major conclusions are summarized as follows:

(1) For various rank coals, the density of the surface functional groups and inorganic corresponds decreased after $ScCO_2$ exposure due to the dissolution and extraction effect of $ScCO_2$;

(2) The interaction with $ScCO_2$ has a minimal influence on pore shape of various rank coals. The micropore SSA and pore volume were increased while values for mesopore decreased;

(3) The combined effects of surface property and pore structure alterations lead to a higher $CO_2$ adsorption capacity at a low pressure but a lower $CO_2$ adsorption capacity at a high pressure;

(4) The adsorption isotherms of the raw and $ScCO_2$-treated coals can be well described by Langmuir model. The monolayer capacity of the various rank coals decreased after exposure to $ScCO_2$, indicating an elimination of the adsorption sites with $ScCO_2$ exposure.

Finally, it can be concluded from the results of this study that $ScCO_2$ exposure has a distinct influence on the surface property, pore structure, and high-pressure $CO_2$ adsorption capacity of the coals, irrespective of its rank.

**Author Contributions:** Z.Z. and Z.L. conceived and designed the experiments; Z.L. performed the experiments; Z.L. and T.W. analyzed the data; Z.Z. and X.L. contributed materials and analysis tools; Z.L. and X.D. wrote the paper.

**Funding:** This research was funded by the National Natural Science Foundation of China, grant number 51674047 and 51611140122, National Science Fund for Distinguished Young Scholars, grant number 51625401 and Special Youth Project of Science and Technology Innovation Enterprise Capital of China Coal Technology Engineering Group Co., Ltd., grant number 2018-2-QN016.

**Acknowledgments:** This study is supported by the National Natural Science Foundation of China (Grant No. 51674047 and 51611140122), National Science Fund for Distinguished Young Scholars (Grant No. 51625401), and Special Youth Project of Science and Technology Innovation Enterprise Capital of China Coal Technology Engineering Group Co., Ltd. (Grant No. 2018-2-QN016).

**Conflicts of Interest:** The authors declare no conflict of interest.

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
