# Peer review of "Supercritical CO2 Exposure-Induced Surface Property, Pore Structure, and Adsorption Capacity Alterations in Various Rank Coals"

_energies, doi:10.3390/en12173294_

Round 1
Reviewer 1 Report
It was a fine and well written paper. The objective of the study was clear and the results were well presented. A few comments from the reviewer are summarized as follows,
line 142, sample weight should be change of sample weight line 227, authors reported the values for DL coal, ZL coal and NM coal reduced slightly, however, they were increased instead as shown in Table 2 line 235, typo,"altercations" > alterations line 284, typo, "-terated: > treated line 318-319, the authors claimed that CO2 adsorption capacity is affected by the combination fo ScCO2 and the seam water. However, the authors did not present clear evidence to demonstate the factor of moisture to the adsorption It would be nice that if authors extended the adsorption measurement of ScCO2 to a much higher pressure, say above 5 Mpa, or even close to 16 Mpa in which that the coal samples have been exposed to. Then the adsorption isotherm may belong to other types.Author Response
Point 1: Line 142, sample weight should be change of sample weight line; line 235, typo, "altercations" > alterations; line 284, typo, "-terated: > treated.
Response 1: Thank you for your comment. The spells have been checked and corrected in line 142, line 235 and line 284.
Point 2: Line 227, authors reported the values for DL coal, ZL coal and NM coal reduced slightly, however, they were increased instead as shown in Table 2.
Response 2: Thank you for your comment. “the values for DL coal, ZL coal and NM coal reduced slightly” has been corrected as: “the values for DL coal, ZL coal and NM coal increased slightly”.
Point 3: Line 318-319, the authors claimed that CO2 adsorption capacity is affected by the combination of ScCO2 and the seam water. However, the authors did not present clear evidence to demonstrate the factor of moisture to the adsorption. It would be nice that if authors extended the adsorption measurement of ScCO2 to a much higher pressure, say above 5 MPa, or even close to 16 MPa in which that the coal samples have been exposed to. Then the adsorption isotherm may belong to other types.
Response 3: Thank you for your comment. In order to eliminate the effect of moisture on the CO2 adsorption, the samples were dried before interact with ScCO2. In our further research, the moisture coal samples will be used. It is evident from reference [27] that if the adsorption measurement of CO2 to a much higher pressure, the adsorption isotherm may belong to other types. But the pressure for the IGA system is limited to 1.8 MPa, where CO2 show monolayer adsorption. Our further research will extend this topic to higher adsorption pressure to better simulate seam conditions.
Reviewer 2 Report
I commend the authors for rigorous testing, analysis and reporting of supercritical CO2 flooding experiments on coal samples of varying quality. I found the article insightful and interesting. I have some questions, comments and general observations which I have listed which I hope can help improve the manuscript before it can be published.
There are significant language issues within the manuscript. These include grammatical issues, spelling errors, as well as contextual issues. I cannot highlight all such instances that I identified since the list of such issues will be quite long. Some examples are: Line 25, 121, 148, 234, 284, 288, 318, etc. I would suggest that you review the manuscript thoroughly and redress these and other issues of language/ grammar where applicable. To me, the uniqueness of this work is not very clear when I read the manuscript in its entirety. Specifically, the authors stress that their work shows that supercritical CO2 exposure impacts the surface properties and pore structure of coal. However, my understanding is that other studies in the past have already demonstrated this. These changes would consequently impact the CO2 adsorption capacity as well. I think the authors should reframe their discussion, both in the introduction as well as the conclusions section to highlight how their observations/ results specifically impact our understanding of CO2 adsorption in coal in ways that are at least an incremental advancement over what is already reported widely by others. For experimental pressure and temperature control, authors suggest that it corresponds with coal seems at ~ 800m depth (Page 4). I would suggest adding a reference which validates that the conditions match expected reservoir conditions. Before discussing specific sections, which highlight experimental methods (such as section 2.3, FTIR); please introduce what these terms mean. For a reader who is not familiar with these techniques, it can be hard to follow. In a similar vein, abbreviations when first used within the manuscript should also be defined for reader’s reference. As an example, Line 134, authors introduce SSA, but it is only defined in the abstract. Consider defining it along with first use within the manuscript as well. Another example is term PSD used in Line 254. For BET/ BJH models, consider providing some background reference or some discussion to orient the readers. Similarly, for pore width measurements, provide some details on model used and calculations. The authors suggest that amount of absorptive pore increases due to combined effect of amount of micropore expansion and the reduction in mesopore volume. However, there is an order of magnitude difference between them and I would assume a similar impact on total volume of absorptive pore as well. Therefore, some basic volumetric calculations/ modeling will be desirable to understand what the relative impacts are. As is, the assertion is merely speculative. From the pore size distribution analysis (Fig 8), it would seem that the mesopore distributions are very similar for GY and DT coal. This is particularly true at >5 nm width. Can the authors suggest any reasons for this? It is interesting that these samples also show similar FTIR spectra for treated samples. IT would be useful if authors could shed some light on said similarities. The authors state that they use the NLDFT method to solve for micropore size distribution. Reader would logically expect some discussion on the method and applicable size ranges, etc. Line 291, authors suggest that the extraction of functional groups on the coal surface by supercritical CO2 could be causing a drop in CO2 adsorption capacity at higher pressures. Can there be some quantitative analysis to see whether the impact can be tied back to the cause that the authors suggest? The reason I ask this is because just looking at the isotherms, it is hard to see the causative relationship. Loss of functional groups is most severe for coal but the decline in adsorption at higher pressures does not seem to be that severe in a relative sense. Can the authors comment on the Langmuir fits and the uniqueness of these fits? Is it possible to get good fits with significantly different nm/ B values? Under conclusions #3, the authors state that SSA and pore volume of micropores increase at the expense of mesopores. Is there a volumetric equivalence here? If not, the statement is misleading and should be modified. The transition in behavior based on the Langmuir isotherms seems to occur at ~1 MPa. Considering downhole conditions, shouldn’t this lead us to conclude that CO2 adsorption always drops with supercritical CO2 flooding? If true, the authors should discuss this further.Author Response
Point 1: There are significant language issues within the manuscript. These include grammatical issues, spelling errors, as well as contextual issues. I cannot highlight all such instances that I identified since the list of such issues will be quite long. Some examples are: Line 25, 121, 148, 234, 284, 288, 318, etc. I would suggest that you review the manuscript thoroughly and redress these and other issues of language/ grammar where applicable.
Response 1: Thank you for your comment. We have reviewed the manuscript thoroughly and correct the grammatical issues, spelling errors and contextual issues in line 25, 121, 148, 234, 284, 288, 318, etc.
Point 2: To me, the uniqueness of this work is not very clear when I read the manuscript in its entirety. Specifically, the authors stress that their work shows that supercritical CO2 exposure impacts the surface properties and pore structure of coal. However, my understanding is that other studies in the past have already demonstrated this. These changes would consequently impact the CO2 adsorption capacity as well. I think the authors should reframe their discussion, both in the introduction as well as the conclusions section to highlight how their observations/ results specifically impact our understanding of CO2 adsorption in coal in ways that are at least an incremental advancement over what is already reported widely by others.
Response 2: Thank you for your comment. It is evident that many previous studies have demonstrated that supercritical CO2 exposure impacts the surface properties and pore structure of coal. But the experimental methods (pressure, temperature, saturation time) varied among each study. In addition, the coal samples were collected from different deposits, which may result in different CO2-coal interaction. The aim of this study is to investigate the effects of supercritical CO2 exposure on the surface properties pore structure, as well as the corresponding adsorption capacity of various coal ranks. On the basis of your comment, we have reframed our introductions, discussions as well as conditions to highlight the results of this study. For example, in line 77, line 328, etc.
Point 3: For experimental pressure and temperature control, authors suggest that it corresponds with coal seems at ~ 800m depth (Page 4). I would suggest adding a reference which validates that the conditions match expected reservoir conditions.
Response 3: Thank you for your comment. Reference [17] have been added in line 109.
Point 4: Before discussing specific sections, which highlight experimental methods (such as section 2.3, FTIR); please introduce what these terms mean. For a reader who is not familiar with these techniques, it can be hard to follow. In a similar vein, abbreviations when first used within the manuscript should also be defined for reader’s reference. As an example, Line 134, authors introduce SSA, but it is only defined in the abstract. Consider defining it along with first use within the manuscript as well. Another example is term PSD used in Line 254. For BET/ BJH models, consider providing some background reference or some discussion to orient the readers. Similarly, for pore width measurements, provide some details on model used and calculations.
Response 4: Thank you for your comment. We have checked the experimental methods and abbreviations and defined them when first used within the manuscript. (1) Line 81, Fourier Transform infrared spectroscopy (FTIR); (2) Line 184, specific surface area (SSA), pore size distribution (PSD). Reference [27] have been added in Line 217.
Point 5: The authors suggest that amount of absorptive pore increases due to combined effect of amount of micropore expansion and the reduction in mesopore volume. However, there is an order of magnitude difference between them and I would assume a similar impact on total volume of absorptive pore as well. Therefore, some basic volumetric calculations/modelling will be desirable to understand what the relative impacts are. As is, the assertion is merely speculative.
Response 5: Thank you for your comment. Although there is an order of magnitude difference between the micropore expansion and mesopore volume reduction as determined by CO2 and N2, the low-pressure N2 adsorption may largely underestimate the mesopore volume due to shrinkage of the coal matrix and the active diffusion effect. Actually, mesopores play a significant role in CO2 adsorption, but it is hard to determine the relative impacts due to the complexity of CO2 adsorption on coal. But it is clear that the adsorption mainly occurs in micropores at a lower pressure, and then translate to mesopore at a higher pressure.
Point 6: From the pore size distribution analysis (Fig 8), it would seem that the mesopore distributions are very similar for GY and DT coal. This is particularly true at >5 nm width. Can the authors suggest any reasons for this? It is interesting that these samples also show similar FTIR spectra for treated samples. It would be useful if authors could shed some light on said similarities.
Response 6: Thank you for your comment. It is evident from Figure 8 that the pore size distributions are very similar for GY and DT coal at >5 nm, which may due to that the GY and DT coal have similar rank. The analyses have been added in the manuscript (line 260). Also, the major peak of FTIR spectra for the samples were similar, indicating the governing surface functional groups on various coals were the same type.
Point 7: The authors state that they use the NLDFT method to solve for micropore size distribution. Reader would logically expect some discussion on the method and applicable size ranges, etc.
Response 7: Thank you for your comment. We have checked the experimental methods and abbreviations and defined them when first used within the manuscript. (1) Line 81, Fourier Transform infrared spectroscopy (FTIR); (2) Line 184, specific surface area (SSA), pore size distribution (PSD). Reference [27] have been added in Line 217.
Point 8: The authors state that they use the NLDFT method to solve for micropore size distribution. Reader would logically expect some discussion on the method and applicable size ranges, etc.
Response 8: Thank you for your comment. The discussion on NLDFT method have been added in Line 266~268. “The micropore size distribution of the raw and ScCO2-treated coal samples was determined by nonlocal density functional theory (NLDFT), an effective method to calculate the pore size from 0.5-1 nm in the micropore range, and the results are shown in Figure 9”.
Point 9: Line 291, authors suggest that the extraction of functional groups on the coal surface by supercritical CO2 could be causing a drop in CO2 adsorption capacity at higher pressures. Can there be some quantitative analysis to see whether the impact can be tied back to the cause that the authors suggest? The reason I ask this is because just looking at the isotherms, it is hard to see the causative relationship. Loss of functional groups is most severe for coal but the decline in adsorption at higher pressures does not seem to be that severe in a relative sense.
Response 9: Thank you for your comment. Due to that FTIR can only provide qualitative information on the surface functional groups alteration, it is hard to conduct any quantitative analysis about its impact on adsorption. But it is evident that the loss of functional groups can decline the adsorption at higher pressures. In our further study, semi-quantification analysis will be conducted to demonstrate this.
Point 10: Can the authors comment on the Langmuir fits and the uniqueness of these fits? Is it possible to get good fits with significantly different nm/ B values?
Response 10: Thank you for your comment. The Langmuir fits were calculated by Origin Lab software in this study. The comments on the Langmuir fits were added in Line 302. “The fitting parameters for the isotherms are determined by Origin Lab software and the best-fit results are summarize in Table 3.”
Point 11: Under conclusions #3, the authors state that SSA and pore volume of micropores increase at the expense of mesopores. Is there a volumetric equivalence here? If not, the statement is misleading and should be modified.
Response 11: Thank you for your comment. There is not volumetric equivalence between the micropores increase and mesopore decrease. This statement has been modified in the manuscript. “The interaction with ScCO2 has a minimal influence on pore shape of various rank coals. However, the micropore SSA and pore volume were increased while the values of mesopore were decreased;”
Point 12: The transition in behavior based on the Langmuir isotherms seems to occur at ~1 MPa. Considering downhole conditions, shouldn’t this lead us to conclude that CO2 adsorption always drops with supercritical CO2 flooding? If true, the authors should discuss this further.
Response 12: Thank you for your comment. The pressure for the IGA system is limited to 1.8 MPa, where CO2 show monolayer adsorption and can obtain well fits by Langmuir model. The results here can only predict the monolayer adsorption behavior. In the seam conditions, it cannot conclude that CO2 adsorption always drops with supercritical CO2 flooding, due to that the adsorption of CO2 would translate to multilayer at a higher pressure, especially when CO2 translate to its supercritical state. This will be one of the main topics of our further studies.